

# Spatial-Temporal Clustering of Tornadoes

Bruce D. Malamud[1], Donald L. Turcotte[2]

[1]Department of Geography, King's College London, London, WC2R 2LS, UK
[2]Department of Geology, University of California Davis, CA, 95616 USA

*Correspondence to*: Bruce D. Malamud (bruce.malamud@kcl.ac.uk)

**Abstract.** The standard measure of the intensity of a tornado is the Fujita or Enhanced Fujita scale, which is based qualitatively on the damage caused by a tornado. An alternative measure of tornado intensity is the tornado path length, L. Here we examine the spatial-temporal clustering of severe tornadoes, which we define as having path lengths L ≥ 10 km. Of particular concern are tornado outbreaks when a large number of severe tornadoes occur in a day in a restricted region. We apply a spatial-

temporal clustering analysis developed for earthquakes. We take all pairs of severe tornadoes in an outbreak, and for each pair plot the spatial lag (distance between touchdown points) against the temporal lag (time between touchdown points). We test our approach applying our analysis to the intense tornado outbreaks in the central United States on 26 and 27 April 2011, which resulted in over 300 fatalities. The patterns of spatial-temporal lag correlations that we obtain for the two days are strikingly different. On 26 April there were 45 severe tornadoes and our clustering analysis is dominated by a complex

sequence of linear features. We associate each linear feature with multiple tornadoes generated by one discrete supercell thunderstorm. On 27 April there were 64 severe tornadoes and our clustering analysis is predominantly random with virtually no embedded linear patterns. We associate this pattern with a widespread complex of interacting supercell thunderstorms without long well defined paths of movement.

## 1 Introduction

It is the purpose of this paper to apply to tornadoes a methodology for spatial-temporal clustering analysis developed by Zaliapin *et al.* (2008) for seismicity. Their methodology considers the times of occurrence and locations of point events. All pairs of events are considered and the spatial lag (distance between a pair of events) $d$ is plotted against the temporal lag (the time difference between the pair of events) $\tau$.

The methodology was developed to decluster earthquake aftershocks from background seismicity. All earthquakes have aftershock sequences; the aftershocks are clusters close to main shocks in both time and space. However, background seismicity (other main shocks) will also occur in the region and time interval that the aftershocks occur. It is important to separate the aftershocks from the background seismicity in order to study the statistics of the aftershocks. Zaliapin *et al.* (2008) demonstrated that plots of spatial lag versus temporal lag clearly separated the two groups of earthquakes.




In this paper, we will consider the time and place of the touchdown of a tornado as a point event. Our studies will be concentrated on tornado outbreaks. An outbreak is a sequence of spatially correlated tornadoes that occur in a relatively short period of time, typically a day. We will consider several examples and will give maps of the tornado touchdown points as well as a clustering analysis of the dependence of spatial lag $d_{ij}$ between the touchdowns of two tornadoes on the temporal lag $\tau_{ij}$

between the touchdown times of the same two tornadoes. We will find significant variability in the clustering analysis patterns. A specific example of a tornado outbreak is a sequence of tornadoes produced by a single convective cell. For this limiting case, the dependence of $d_{ij}$ on $\tau_{ij}$ is approximately linear, and the slope is the velocity of the convective cell.

To illustrate this clustering analysis methodology applied to tornadoes, we consider a sequence of four point events that occur

at successive times $t_1$, $t_2$, $t_3$, $t_4$ and two dimensional locations $(x_1, y_1)$, $(x_2, y_2)$, $(x_3, y_3)$, $(x_4, y_4)$, as illustrated in **Figure 1**. The temporal lags (time differences) are $\tau_{12} = t_2-t_1$, $\tau_{13} = t_3-t_1$, $\tau_{14} = t_4-t_1$, $\tau_{23} = t_3-t_2$, $\tau_{24} = t_4-t_2$, and $\tau_{34} = t_4-t_3$. The corresponding spatial lags (spatial separations) are $d_{12} = [(x_2-x_1)^2 + (y_2-y_1)^2]^{0.5}$ and $d_{13}$, $d_{14}$, $d_{23}$, $d_{24}$, and $d_{34}$ determined in a similar way. The time of occurrences of the temporal lags $\tau$ for our four point events are illustrated in **Figure 1a** and the two-dimensional locations of the spatial lags $d$ in **Figure 1b**. The dependence of the spatial lags $d$ on the temporal lags $\tau$ are given in **Figure**

**1c**. In this paper, we will give the dependence of spatial on temporal lags for pairs of tornado touchdowns.

Studies of the statistics of tornadoes are limited by the problems associated with the quantification of tornado intensity. Ideally, tornado intensities would be based on wind speed measurements. However, as noted by Doswell *et al.* (2009), high resolution Doppler measurements of wind velocities in tornadoes are not possible at this time. Currently, the Fujita and Enhanced Fujita

scales are the standard measure of tornado intensities (Fujita, 1971). Tornadoes are classified on a scale of F0 to F6 based on a qualitative measure of damage. An alternative measure of tornado intensity is the tornado path length, *L*. In the United States, the NOAA (2015) Storm Prediction Center Severe Weather Database (SPC—SWD) provides Fujita scale values and path lengths for tornadoes. Brooks (2004) has provided a detailed study of the statistical correlations between the Fujita scale intensities and the path length. Malamud and Turcotte (2012) extended these studies and defined a severe tornado to be a

tornado with path length $L \geq 10$ km. In the studies reported in this paper, we will retain this definition and consider only tornadoes with $L \geq 10$ km. A path length of 10 km corresponds roughly to a F2 tornado (Malamud and Turcotte, 2012). Most severe tornadoes are generated by supercell thunderstorms (Doswell *et al.*, 1993). A supercell thunderstorm can be defined as a long-lived (> 1 hr) thunderstorm, with a high degree of spatial correlation between its mesocyclone (the vortex of air within the storm) and updraft (Davies-Jones *et al.* 2001).


The objective of this paper is to study the clustering statistics of tornado outbreaks. However, it must be recognized that the definition of a tornado outbreak is somewhat arbitrary (Mercer *et al.*, 2009). Ideally, the definition of a tornado outbreak would be the occurrence of multiple tornadoes within a particular synoptic-scale weather system (Glickman, 2000). Galway (1977)



classified tornado outbreaks into three types: (i) A local outbreak with a radius less than 1000 miles (1609 km); (ii) a progressive outbreak moving from west to east in time, and (iii) a line outbreak associated with a single moving supercell thunderstorm. Unfortunately, the NOAA (2015) NWS—SPC database does not associate individual tornadoes with a specific tornado outbreak, using any of these three (or other) classifications.

There is a strong diurnal variability in tornado occurrence associated with solar heating. For these reasons, Doswell *et al.* (2006) defined a tornado outbreak to include all tornadoes in the continental USA in a convective day, i.e. the 24 hr period from 12:00 UTC (Coordinated Universal Time) of a given day to 12:00 UTC of the following day. The Severe Weather Database that we use in our analyses list most tornadoes in Central Standard Time (CST), so we will consider tornadoes in a
convective day as 06:00–06:00 CST. However, consistent with the studies of severe tornado outbreaks given by Malamud and Turcotte (2012), we will consider a severe tornado outbreak to include only those tornadoes with path lengths $L \geq 10$ km.

## 2 Clustering analysis of Tornadoes

To illustrate our clustering analysis methodology for tornadoes, we will first consider the intense tornado outbreaks in the Central United States on 26 and 27 April 2011. The tornado outbreaks in the spring of 2011 have been discussed in detail by
Doswell *et al.* (2012). They concluded that ideal conditions for severe tornado outbreaks occurred during the last two weeks of April 2011, and that the supercell thunderstorms responsible for the tornadoes were generated by a sequence of extra-tropical cyclones. In this paper, we focus our attention on the 2011 outbreaks that occurred on 26 and 27 April. Although these outbreaks were certainly related to the same synoptic scale weather pattern, we will treat the two outbreaks separately for our statistical studies. We will consider severe ($L \geq 10$ km) tornadoes on convective days: (i) 06:00 CST 26 April 2011 to 06:00
CST 27 April 2011 (i.e., a convective day equivalent to 12:00 UTC 26 April 2011 to 12:00 UTC 27 April 2011), (ii) 06:00 CST 27 April 2011 to 06:00 CST 28 April 2011.

In **Figure 2** we give touchdown times $t$ and path lengths $L$ for the 45 severe ($L \geq 10$ km) tornadoes that occurred on 26 April 2011 (convective day, 06:00 CST to 06:00 CST of the following day) and for the 64 severe tornadoes that occurred on 27 April
2011 (convective day). In Malamud and Turcotte (2012) we suggested that a quantitative measure of the strength of a severe tornado outbreak is the total path length $L_D$ of all severe ($L \geq 10$ km) tornadoes in a convective day in the continental USA. By this measure the strongest tornado outbreak during the 60 year period 1954–2013 was on 3 April 1974 (convective day) with 105 severe ($L \geq 10$ km) tornadoes and a total tornado path length $L_D = 3852$ km. For the two outbreaks illustrated in **Figure 2**, the outbreak on 26 April 2011 with 45 severe tornadoes had a total tornado path length $L_D = 1239$ km, the 5[th] strongest outbreak
during this same 60 year period, 1954–2013. The outbreak on 27 April 2011 with 64 severe tornadoes had a total path length $L_D = 2815$ km, the 2[nd] strongest outbreak during this period.



We next consider the spatial distributions of the tornado touchdown points for both the 26 and 27 April 2011 outbreak events. In **Figure 3a** we give a map of the tornado paths of the 45 severe ($L \geq 10$ km) tornadoes that occurred on 26 April 2011 (convective day) and in **Figure 3b**, the 64 severe tornadoes that occurred on 27 April 2011 (convective day). In **Figure 3**, the tornado touchdown are given by symbols and the paths by lines. The symbols for tornado touchdowns are given by shapes and

colours, with combinations defining eight three-hour periods for the initial touchdown times. The lines for each tornado path length illustrate the overall tornado movements. Tornado path lengths vary from 10 km (our lower cut-off for a severe tornado) to 113.3 km (26 April 2011) and 212.4 km (27 April 2011). We will postpone a discussion of the regions A and B that are indicated on **Figure 3a** until a later section. In **Figure 3a**, although there tends to be a southwest–northeast trend to the 26 April 2011 touchdowns, the spatial distribution appears visually to be diffuse. In **Figure 3b**, the southwest-northeast trend of

the 27 April 2011 touchdowns is visually less diffuse than in **Figure 3a**.

We now turn to our clustering analyses of the two tornado outbreaks on the 26 and 27 April 2011. From the times of occurrence given in **Figure 2** and the spatial locations of tornado touchdowns given in **Figure 3a** and **3b**, we obtain the temporal and spatial lags using the method illustrated in **Figure 1**. In **Figure 4a** we give the spatial-temporal lag correlations of all pairs of

the 45 severe ($L \geq 10$ km) tornado touchdowns that occurred on 26 April 2011 (convective day). The number of pairs are $N_P = 1 + 2 + \ldots + (N_T - 1)$, with $N_T$ the number of tornadoes considered. With $N_T = 45$ tornadoes, we have $N_P = 990$ data points on the plot. There are quite clear near linear trends to the $d$ (spatial lags) vs. $\tau$ (temporal lags) data given in **Figure 4a**, with the spatial lags increasing with the temporal lags.

Consider the spatial-temporal lag correlations associated with a series of tornadoes generated by a single supercell thunderstorm moving at a constant velocity $v_c$. The correlations can be approximated by a near linear trend passing through the origin with the slope giving the velocity $v_c$. This behaviour will be demonstrated in some detail in the next section. This behaviour explains the strong linear trend passing through the origin in **Figure 4a**. We fit a straight line to this trend in **Figure 4a**, and obtain a velocity $v_c = 70$ km hr$^{-1}$. We associate this trend with the southwest-northeast movement of a supercell

thunderstorm as discussed above. Within the cloud of spatial-temporal lag data considered in **Figure 4a**, there appears to be linear trends with slightly different slopes. We associate these differences with tornadoes within one or more supercell thunderstorms that have different velocities $v_c$.

We next turn our attention to one of the near linear trends in **Figure 4a** that does not pass through the origin, indicated by the

rectangular region AB. We return to **Figure 3a**, where in Region A we outlined a spatial cluster of the touchdowns for three severe tornadoes that occurred on 26 April 2011, and in Region B, a spatial cluster of the touchdowns for 14 severe tornadoes. In the rectangular region AB given in **Figure 4a** there are 51 data points, of which 42 (82%) of them represent all of the pairs of tornado touchdowns between the two regions A and B in **Figure 3a**, with none of the data points in box AB associated with pairs of tornadoes within Region A or pairs of tornadoes in region B. We find that this explanation of correlations between



tornadoes generated by two separate supercell thunderstorms (the spatial regions A and B in **Figure 3a**) provides an explanation for the near-linear trends of spatial and temporal lags observed in **Figure 4a**.

In **Figure 4b**, we give the spatial lag vs. temporal lag for each of the pairs of the 64 severe ($L \geq 10$ km) tornado touchdowns that occurred on 27 April 2011 (convective day). In this case there are $N_P = 2016$ pairs. Comparing **Figure 4b** with **Figure 4a**, there are striking differences. Specifically, in **Figure 4b**, there is no clear near linear trend of the spatial-temporal lag data; whereas, in **Figure 4a**, this linear trend both through the origin and in other spatial-temporal lag regions of the plot is dominant. One potential conclusion is that the generation of tornadoes by several separately defined supercell thunderstorms is absent on 27 April 2011, and rather, there were many interacting supercell thunderstorms on this day (e.g., Knupp, 2013). The large scatter in data indicates the tornadoes do not occur along well defined spatial paths in **Figure 3b**.

## 3 Interpretation of Results

In order to better understand the implications of our spatial-temporal lag correlations, we consider a simple example. We consider an idealized model of a progressive tornado outbreak generated by a single supercell thunderstorm moving at a constant velocity, $v_c$. In our example, we take a six-hour time window during which a supercell moves at a uniform velocity $v_c = 80$ km hr$^{-1}$ along a linear track with a length of $L_{track} = 480$ km. We assume that five model tornadoes touchdown at random times during the six hour time period. The times $t$ and locations of touchdowns are illustrated in **Figure 5a**. The spatial-temporal lag correlations between the model tornado touchdowns in **Figure 5a** are shown in **Figure 5b**. The spatial lag $d$ is plotted against the temporal lag $\tau$ for each of the ten pairs of tornadoes. The data fall on a straight line that defines the velocity $v_c = 80$ km hr$^{-1}$. Spatial-temporal lag correlations that fall on or close to a straight line going through the origin are indicative of a progressive tornado outbreak. The 26 April 2011 outbreak correlation data given in **Figure 4a** have a strong linear trend well approximated by a velocity $v_c = 70$ km hr$^{-1}$.

As a confirmation of our association given above we consider a progressive tornado outbreak that occurred on 4 April 2011 (convective day). We consider 6 severe ($L \geq 10$ km) tornadoes that occurred between 13:42 and 18:43 CST, which gives 15 pairs of tornadoes. Three severe tornadoes on that day that were spatially distant (>600 km from any of the six tornadoes) were not considered. In **Figure 6**, the spatial lag $d$ is plotted against the temporal lag $\tau$ for each of these 15 pairs of tornado touchdown points. On the basis of the linear correlations given in **Figure 5b**, we compare the data values in **Figure 6** with a least-squares fit to a linear correlation passing through the origin, resulting in a supercell velocity of $v_c = 68.5$ km hr$^{-1}$ (Spearman rank correlation coefficient, $r^2 = 0.92$).

We now return to a discussion of the well-defined linear trends in the spatial-temporal correlation given in **Figure 4a**. The first linear trend, extending from the origin with a slope corresponding to $v_c = 76.0$ km hr$^{-1}$, can be explained as we explained the



similar linear trends in **Figures 5 and 6**. For the second linear trend within the box AB of **Figure 4a**, we determined that these points were the result of spatial-temporal correlations between the tornadoes in boxes A and B in **Figure 3a**. Most of the data points (82%) in box AB in **Figure 4a** were the result of spatial-temporal lag correlations between boxes A and B in **Figure 3a**. The approximately 300 km vertical offset distance at zero time lag in **Figure 4a** between the origin and box AB is approximately the distance between the nearest touchdown locations between Regions A and B in **Figure 3a**.

We next introduce a measure of the combined spatial-temporal separation of pairs of tornado touchdowns, where the spatial-temporal separation $\psi$ is given by:

$$\psi = \tau + \frac{d}{v_c} \tag{1}$$

where, as previously, $\tau$ and $d$ represent, respectively, the temporal and spatial lags between the tornado touch downs, and $v_c$ the average supercell velocity, which we take here to be $v_c = 80$ km hr$^{-1}$. Small values of both temporal and spatial lag, result in small values of the spatial-temporal separation. For example, if the lags between two tornado touchdowns are $\tau = 2$ hr and $d = 160$ km, then the spatial-temporal separation $\psi = (2$ hr$) + (160$ km$)/( 80$ km hr$^{-1}) = 4$ hr. We consider the statistical distribution of the values of $\psi$, by introducing the normalized cumulative probability defined as:

$$P(<\psi) = \frac{N_C(<\psi)}{N_T} \tag{2}$$

with $N_C(<\psi)$ the number of tornado touchdown pairs with spatial-temporal separation values less than $\psi$, and $N_T$ the total number of pairs considered.

In **Figure 7** we plot the normalized cumulative probability P($<\psi$) as a function of the spatial-temporal separations $\psi$. We consider the data for the two tornado outbreaks in the USA on 26 and 27 April 2011 (convective days) and utilize the values given in **Figure 3** for spatial-temporal separations $\psi < 4$ hr. We have not considered data for the 4 April 2011 outbreak given in **Figure 6**, because of the very small number of data points.

We see that the sets of normalized cumulative probability values for the two outbreaks given in **Figure 7** have a very different pattern, one linear and the other a power-law. For the 26 April 2011, our data set consisted of 45 severe tornadoes resulting in $N_P = 990$ pairs of tornado touchdowns of which 245 spatial-temporal separations are illustrated in **Figure 7**. The least-squares best fit linear correlation for the spatial-temporal separations for 26 April 2011, over the range $0.0 < \psi < 4.0$ hr, gives:

$$P(<\psi) = 0.0671\psi - 0.0241 \tag{3}$$

which is in excellent agreement with the data in the range $0.6 < \psi < 4.0$ hr. For the 27 April 2011 outbreak, our data set consisted of 64 severe tornadoes resulting in $N_P = 2016$ pairs of tornado touchdowns of which 330 spatial-temporal separations





are illustrated in **Figure 7**. The least-squares best fit power-law correlation for the spatial-temporal separations for 27 April 2011, over the range $0.0 < \psi < 4.0$ hr, gives::

$$P(<\psi) = 0.011\psi^{1.95} \qquad (4)$$

which is in excellent agreement with the data and has an exponent close to 2.

We now give an explanation for the linear and power-law correlations that we have found. If the tornado touchdowns occur randomly along a path for relatively small values of spatial-temporal separations $\psi$, then a linear correlation of normalized cumulative probability P($<\psi$) with spatial-temporal separation $\psi$ is expected to be a good approximation. In contrast, if the tornado touchdowns occur randomly in both space and time, then it is expected that P($<\psi$) is proportional to $\psi^2$, i.e. the area
of the segment of a circle of possible touchdown locations. The transition from linear to random behaviour indicated by the data in **Figure 7** is consistent with our previous qualitative discussion of the data given in **Figure 4**.

**4 Discussion**

Unlike many other natural hazards, it is difficult to quantify strong tornadoes. For hurricanes there are extensive data on wind speeds and barometric pressures. For floods, flood gauges are a quantitative measure of the flow rate in a river. For earthquakes,
seismographs give measures of shaking intensity. Quantifying volcanic eruptions and landslides is more difficult but volumes of material involved can be estimated. It is not possible to measure reliably the wind speeds or pressure changes in tornadoes. The standard measure of tornado intensity used today is the Fujita (or enhanced Fujita) scale. This scale is based qualitatively on the damage caused by a tornado. An alternative measure of tornado intensity is the tornado path length, *L*.

Malamud and Turcotte (2012) showed that records for the 1990s to present of tornado path lengths are relatively complete for severe tornadoes (defined to be $L \geq 10$ km) in the United States. They also showed that the number-length scaling of severe tornado touchdowns is well approximated by a power-law distribution. Elsner *et al.* (2014) showed that the distribution of daily tornado counts in the United States is also well approximated by a power-law relationship. Malamud and Turcotte (2012) also studied the statistics of recent severe tornado outbreaks. They quantified the strength of a severe tornado outbreak to be
the total tornado path length $L_D$ of the severe tornadoes occurring during a convective day (06:00 CST of a given day, to 06:00 CST of the next day). They showed that the number-length scaling of severe tornado outbreaks is also well approximated by a power-law distribution.

In this paper we consider an alternative statistical measure of a tornado outbreak. We apply a spatial-temporal clustering
analysis developed by Zaliapin *et al.* (2008) for earthquakes. We consider the sequence of severe tornado touchdowns occurring during a convective day to be a sequence of point events in space and time. We consider all pairs of these point



events and plot the spatial lag (i.e., spatial distance between the touchdown points for a pair of events) versus the temporal lag (difference in touchdown times between the same pair of events). In terms of expectations for this data there are two limiting cases:

(1)  *Tornadoes occurring randomly in time and space* during a specified time interval and spatial area. The expected pattern
will depend on the number of events and the shape of the spatial area considered. The resulting plot of spatial lags $d$ versus temporal lags $\tau$ will be highly random with a systematic decrease in density away from the origin. No distinctive patterns will occur. In Eq. (1) we introduced the variable $\psi$ as a measure of the combined spatial-temporal separation of pairs of tornado touchdowns. For the random occurrence of touchdowns in space and time, and relatively small values of $\psi$, we expect that the cumulative number of values of $\psi$ will scale as $\psi^2$. Small values of $\psi$ are used so as to eliminate
edge effects that come in for large $\psi$.

(2)  Tornadoes generated by a single supercell thunderstorm moving on a nearly linear path at a near constant cell velocity $v_c$. In this case, the data points will be approximated by a straight line in $d$ versus $\tau$ space with the slope given by velocity $v_c$. In this case we expect that the cumulative number of values of $\psi$ will scale approximately as $\psi$.

The principal focus of this paper has been the application of a clustering analysis to several tornado outbreaks. It is expected that a small outbreak of severe tornadoes in a given convective day could be associated with a single, or at most a few supercell thunderstorms. An example of the spatial-temporal lag correlations for six severe tornadoes on 4 April 2011 (convective day) is given in **Figure 6**, with a good linear fit as expected, and the gradient of the line representing the cell velocity.

We also applied our clustering analysis to the intense tornado outbreaks in the central United States on 26 and 27 April 2011, with 45 and 64 severe tornadoes occurring, respectively (convective days), and more than 300 fatalities. For each pair of tornadoes on the two separate days, the severe tornado touchdown spatial lags are given as a function of their temporal lags in **Figure 4**. The observed patterns are very different. The results for 26 April 2011 (convective day) in **Figure 4a** are dominated by a complex sequence of linear tracks that we have previously discussed. This pattern is consistent with the movement of a
discrete set of supercell thunder storms moving from the southwest to the northeast at velocities near 70 km hr$^{-1}$. The observed pattern for 27 April 2011 (convective day) given in **Figure 4b** is quite different. It is predominantly random with virtually no embedded linear patterns that can be associated with the movement of individual supercell thunderstorms. This conclusion is confirmed by the strong separation of the data for the two days illustrated in **Figure 7.** One explanation would be a widespread complex of interacting super cell thunderstorms without well-defined long paths of movement.


Based on our spatial-temporal lag results for 26 and 27 April 2011 (**Figure 4**) we believe our clustering analysis approach provides useful quantitative information on the structure of severe tornado outbreaks. Tornado touchdowns are not available during an outbreak, thus these studies can only be carried out retrospectively. A possible extension of our study would be to



obtain the spatial-temporal lag statistics of lightning strikes that occur during the same time period as the tornado outbreaks that we are studying, as both are associated with supercell thunderstorms. These data are available during a tornado outbreak and their analyses merits future studies.

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



**Figure 1**. **Illustration of our clustering analysis methodology. (a) A sequence of four point events occur at times $t_1$, $t_2$, $t_3$, $t_4$. The six temporal lags $\tau_{12}$, $\tau_{13}$, $\tau_{14}$, $\tau_{23}$, $\tau_{24}$, $\tau_{34}$ are shown. (b) The two dimensional locations of the four point events are shown. The six spatial lags, $d_{12}$, $d_{13}$, $d_{14}$, $d_{23}$, $d_{24}$, $d_{34}$ are also shown. (c) The six spatial lags $d_{ij}$ are shown as a function of the corresponding temporal lags $\tau_{ij}$, where $i$ is the first event and $j$ is the second event in time.**





**Figure 2**. **Times of touchdown and path lengths of severe tornadoes ($L \geq 10$ km) that occurred on 26, 27 and 28 April 2011. There were 45 severe tornadoes on 26 April (convective day, 06:00 CST to 06:00 CST the following day) and 64 severe tornadoes on 27 April (convective day). Data obtained from NOAA (2015).**




**Figure 3**. **Touchdown locations of (a) 45 severe ($L \geq 10$ km) tornadoes that occurred on 26 April 2011 (convective day, 06:00 CST–06:00 CST), and (b) 64 severe ($L \geq 10$ km) tornadoes that occurred on 27 April 2011 (convective day). The touchdowns points for each tornado are given by colours and shapes (as given in the legend), representing successive three hour intervals. The tornado path lengths for each tornado are given by thin black lines. In (a) the tornadoes outlined in the regions A and B will be discussed in a later section. Data obtained from NOAA (2015).**






**Figure 4**. **Spatial-temporal lag correlations between the touchdowns for (a) 45 severe ($L \geq 10$ km) tornadoes that occurred on 26 April 2011 (convective day) and (b) 64 severe ($L \geq 10$ km) tornadoes that occurred on 27 April 2011 (convective day). The spatial lag $d$ is plotted against the temporal lag $\tau$ for each of the (a) $N_P = 990$ pairs of tornado touchdowns and (b) $N_P = 2016$ pairs of tornado touchdowns. So that (a) and (b) have the same spatial-temporal limits, 147 (7%) of the 2016 data points for (b) that have large spatial or temporal values are not included. The data points in region AB in (a) are correlations between the spatial-temporal lags for the tornadoes in Region A and Region B in Figure 3a.**






**Figure 5**. **Five model tornado touchdown points located randomly in time during a six hour time window along a linear track. (a) The touchdown positions $x$ along the track are shown as a function of the random time $t$ of occurrence. The model super cell thunderstorm responsible for the tornadoes moves along the track at a velocity $v_c = 80$ km hr$^{-1}$. (b) Spatial-temporal lag correlations between the 5 model tornadoes shown in Figure 5a. The spatial lag $d$ is plotted against the temporal lag $\tau$ for each of the 10 pairs of model tornado touchdown points.**





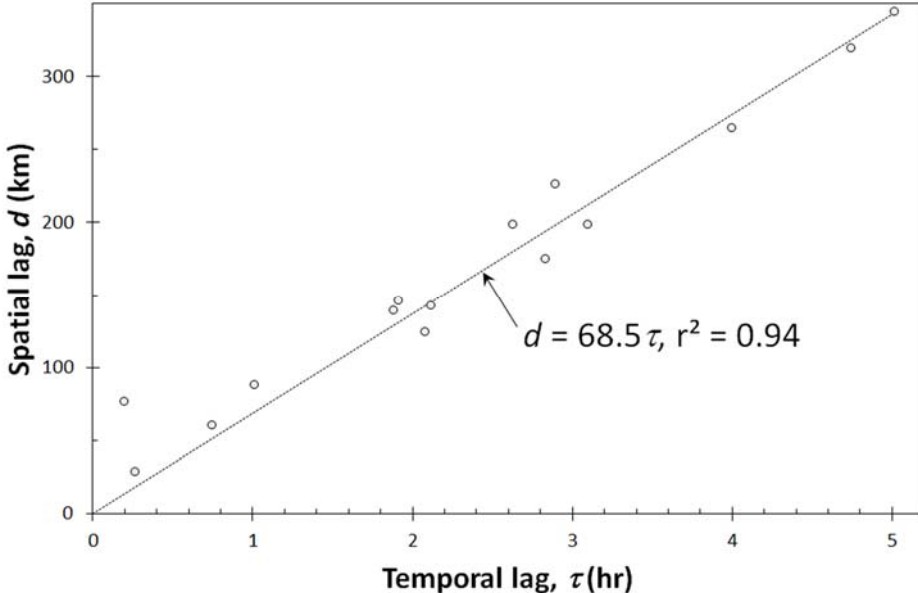

**Figure 6.** **Spatial-temporal lag correlations between 6 severe ($L \geq 10$ km) tornado touchdowns that occurred on 4 April 2011 (convective day), with touchdown times from 13:42 to 18:43 CST. The spatial lag $d$ is plotted against the temporal lag $\tau$ for each of the 15 pairs of tornado touchdown points. The straight-line fit to the data passing through the origin gives a velocity $v_c = 68.5$ km hr$^{-1}$.**







**Figure 7**. **Normalized cumulative probability P(<$\psi$) of spatial-temporal separations $\psi$ between pairs of severe tornado (path length $L \geq 10$ km) touchdowns during tornado outbreaks in the USA on 26 and 27 April 2011 (convective days) (see Figures 2 and 3). Cumulative probabilities are given for spatial-temporal separations $\psi < 4$ hr. Also shown are the least-square best-fit linear (blue dashed line) and power-law (red dashed line) to the data shown in this figure for 26 and 27 April 2011, respectively.**