# Peer review of "Spatial-Temporal Clustering of Tornadoes"

_Natural Hazards and Earth System Sciences, 2016_

## Referee Comment (RC1) · J. Elsner (Referee) · 30 Mar 2016

Review of Malamud and Turcotte. Spatial-temporal clustering of tornadoes.

The authors illustrate a method for analyzing space-time clustering of tornadoes that was originally developed for earthquakes. They do this using the dramatic outbreaks of 26-27 April 2011 and the less dramatic outbreak of 4 April 2011. The work provides convincing evidence that the clustering approach results in useful quantitative information on the structure of severe tornado outbreaks. The paper is well written and straight forward. I recommend publication subject to consideration of the following relatively minor points.

1. The physical rationale for the methodology appropriate for earthquakes might not be appropriate for tornadoes. Perhaps the authors can comment on this? All earthquakes

have aftershocks but not all tornadoes have after events. That is earthquakes can directly cause aftershocks. Tornadoes do not directly cause other tornadoes.

2. The authors note the difficulty in defining a tornado outbreak. Recent work by Elsner et al. (2014) provide a way to do this using spatial distances and a clustering algorithm. Perhaps the authors can take a look at this earlier work and put their work in context? Elsner et al. (2014) The increasing efficiency of tornado days in the United States. Climate Dynamics DOI 10.1007/s00382-014-2277-3.

3. Line 4 on page 2. Define the subscripts (e.g., i is the first event and j is the subsequent event).

4. Line 20 on page 2. I assume the maximum EF rating is 5 not 6.

5. Comment: It seems that the method quantifies an underlying group velocity of multiple tornado events.

---

## Referee Comment (RC2) · H. Brooks (Referee) · 31 Mar 2016

The technique may have some value in some application, but the physical interpretation is fundamentally wrong. The 26 April storms were characterized by quasi-linear structure and the 27 April storms were predominantly discrete supercells. The 4 April case used as proof of "supercell" is actually a derecho. The authors lean on the reverse of that being true. The use of a single supercell as an exemplar is oversimplistic and misleading. If, instead, multiple parallel supercells are assumed, the linear relationship between lag-d and lag-t breaks down. In fact, for the three cases shown, the linear relationship on the plot is indicative of tornadoes associated with linear convective modes, not supercells.

1. p. 1, last paragraph: It's not obvious to me that there's a great physical similarity

between the earthquake/aftershock distribution in time and space and tornado outbreaks. Can you offer insight into why one phenomenon (earthquake sequences) that don't seem to me to have a lot of sense of direction provides a model for one (tornado outbreaks) that has a strong sense of direction?

2. p. 2, line 20: Given that these tornadoes all occurred during the Enhanced Fujita scale era, they should be labelled "EF", not "F". Also, the highest value on the scale is EF5, not EF6.

3. p. 4, lines 25-27 and p. 5, lines 8-10: The "potential conclusion" on p. 5 is not consistent with the radar presentation. If you had said that the generation of tornadoes by several separately defined supercells is absent on 26 April, I'd believe it. In the Knupp et al. (2013) reference, they talk about tornadoes associated with a quasi-linear convective system (QLCS) for the early morning tornadoes in Alabama on 27 April (prior to 12 UTC, so associated with the 26 April convective day) and discrete supercell tornado formation for the afternoon/evening of 27 April. The 26th April storms are much less likely to be supercellular than the 27th April storms. This is reflected in the damage ratings associated with the >10 km path tornadoes for the two days.

EF 26th 27th

EF0 4 3

EF1 20 19

EF2 17 12

EF3 5 15

EF4 0 11

EF5 0 4

The 1240 UTC 27 April tornado in Georgia is associated with the end of 26 April outbreak. There are 4 tornadoes in Alabama (1 EF0, 3 EF1) between 16 and 17 UTC

on the 27th that Knupp et al. associate with a small QLCS. The heart of the 27 April outbreak begins after 18 UTC and is unequivocally from supercells, mostly discrete, as opposed to the QLCSs of the 26th (Knupp et al. 2013). The radar loops are available at http://www.srh.noaa.gov/bmx/?n=event_04272011

4. p. 5, lines 23-29: The 4 April 2011 event was not super-cellular. It was a derecho event with some embedded tornadoes (http://www.spc.noaa.gov/misc/AbtDerechos/casepages/apr042011page.htm). It appears that the technique proposed here with straight lines on the spatial/temporal lag chart is identifying tornado events associated with linear convection, not supercells.

5. Figure 5: A single supercell is not an appropriate model for a tornado outbreak such as 27 April 2011 where multiple supercells were producing tornadoes. I've done a simple exercise with 4 supercells on parallel paths. Each produces a tornado once per hour 100 units to the east and 10 units to the north of the previous tornado. The 2nd (3rd) [4th] supercell is 50 (100) [150] units north of the first. When I calculate the lag d and lag t, rather than getting a straight line, I get the plot below. I'm confident that if I made a slightly more sophisticated model with the tornadoes not occurring at the exact same time and not due north of each other, and with not identical forward speeds, the plot would fill in much like Figure 4b.

It's probable that, if you created this plot and looked at tornadoes associated with a single supercell separately, you'd see a series of lines. This requires identifying the individual supercell (radar loops would help) or cases such as Palm Sunday 1965, the Superoutbreak in 1974, and May 3 1999 have already had that done. For 27 April, a plot of the tornadoes from Mississippi to northeast Georgia associated with the storm that produced the Tuscaloosa tornado would be illustrative.
* * *
[Figure]

[Figure]

**Fig. 1.** Simple model-4 supercells

---

## Referee Comment (RC3) · H. Brooks (Referee) · 1 Apr 2016

As an example from real data, I went back and got the starting locations for the >10 km tornadoes on 3 May 1999 in the NWSFO Norman forecast area (http://www.srh.noaa.gov/oun/?n=events-19990503-stormdata). I created the plot as in the paper. Combinations associated with a single supercell are shown in red and the combinations of tornadoes from different supercells are shown in black. The individual supercell tornadoes come very close to being on straight lines, although the difference in storm motions leads to slightly different lines between the supercells. The between supercell combinations are scattered around.

[Figure]

[Figure]

**Fig. 1.** Lag-t Lag-d for 3 May 1999 supercells in central Oklahoma

---

## Author Comment (AC1) · 30 Jul 2016

We thank both reviewers for their detailed comments, and refer them to the attached supplement where we have addressed their comments in detail.

Please also note the supplement to this comment: http://www.nat-hazards-earth-syst-sci-discuss.net/nhess-2016-71/nhess-2016-71-AC1-supplement.pdf

---

## Author Comment (AC2) · 30 Jul 2016

**Reply to Referee Comments**

We thank the two referees James Elsner and Harold Brooks for their stimulating and helpful comments on the manuscript nhess-2016-71 (Malamud and Turcotte, 2016). We have done further analyses to address some of the concerns raised, along with expanding the text in certain locations and correcting some typos; we believe the result is a substantially improved paper. We have divided our response below into responding to each editor separately. Any page numbers referred to below are the original NHESS manuscript submitted.

**I. REFEREE James Elsner (JE) COMMENTS AND AUTHOR RESPONSE**

*JE Overall Comment "The authors illustrate a method for analyzing space-time clustering of tornadoes that was originally developed for earthquakes. They do this using the dramatic outbreaks of 26-27 April 2011 and the less dramatic outbreak of 4 April 2011. The work provides convincing evidence that the clustering approach results in useful quantitative information on the structure of severe tornado outbreaks. The paper is well written and straight forward. I recommend publication subject to consideration of the following relatively minor points."*

*[JE 1] "The physical rationale for the methodology appropriate for earthquakes might not be appropriate for tornadoes. Perhaps the authors can comment on this? All earthquakes have aftershocks but not all tornadoes have after events. That is earthquakes can directly cause aftershocks. Tornadoes do not directly cause other tornadoes."*
[Reply to JE 1]: We introduce the application to earthquakes since this was where the method we use was introduced. In both earthquakes and tornadoes there is clustering, but the type differs considerably. In earthquakes individual clusters are centred whereas in tornadoes linear trends often occur. In our revised manuscript, we have modified our discussion to address the comment. See also reply to [HB 1].

*[JE 2] "The authors note the difficulty in defining a tornado outbreak. Recent work by Elsner et al. (2014) provide a way to do this using spatial distances and a clustering algorithm. Perhaps the authors can take a look at this earlier work and put their work in context? Elsner et al. (2014) The increasing efficiency of tornado days in the United States. Climate Dynamics DOI 10.1007/s00382-014-2277-3."*
[Reply to JE 2]: In our revised manuscript, we include a reference to this work and discuss how the methodology applied there differs from our approach, mainly in that we are considering both time and space and searching for near-linear features.

*[JE 3] "Line 4 on page 2. Define the subscripts (e.g., i is the first event and j is the subsequent event)."*
[Reply to JE 3]: This has now been clarified in our revised manuscript.

*[JE 4] "Line 20 on page 2. I assume the maximum EF rating is 5 not 6."*
[Reply to JE 4]: This has been changed in our revised manuscript.

*[JE 5] "Comment: It seems that the method quantifies an underlying group velocity of multiple tornado events."*
[Reply to JE 5]: We agree.

**II. REFEREE Harold Brooks (HB) COMMENTS AND AUTHOR RESPONSE**

*[HB Overall Comment] "The technique may have some value in some application, but the physical interpretation is fundamentally wrong. The 26 April storms were characterized by quasi-linear structure and the 27 April storms were predominantly discrete supercells. The 4 April case used as proof of "supercell" is actually a derecho. The authors lean on the reverse of that being true. The use of a single supercell as an exemplar is oversimplistic and misleading. If, instead, multiple parallel supercells are assumed, the linear relationship between lag-d and lag-t breaks down. In fact, for the three cases shown, the linear relationship on the plot is indicative of tornadoes associated with linear convective modes, not supercells."*

*[HB 1] p. 1, last paragraph: It's not obvious to me that there's a great physical similarity between the earthquake/aftershock distribution in time and space and tornado outbreaks. Can you offer insight into why one phenomenon (earthquake sequences)*

*that don't seem to me to have a lot of sense of direction provides a model for one (tornado outbreaks) that has a strong sense of direction?*

[Reply to HB 1]: Background earthquakes are randomly distributed in space and time; however, aftershocks are clustered in space and time. The technique we utilize was developed to identify aftershock clusters. Tornadoes are also clustered in space and time, where one extreme is a quasi-linear cluster associated with a single supercell thunderstorm. Another extreme is a large number of supercell thunderstorms which give an approximately uniform distribution in space and time. We have modified our discussion to more clearly explain these points.

*[HB 2] p. 2, line 20: Given that these tornadoes all occurred during the Enhanced Fujita scale era, they should be labelled "EF", not "F". Also, the highest value on the scale is EF5, not EF6.*

[Reply to HB 2]: We have changed F to EF and changed the largest value to EF5.

*[HB 3] p. 4, lines 25-27 and p. 5, lines 8-10: The "potential conclusion" on p. 5 is not consistent with the radar presentation. If you had said that the generation of tornadoes by several separately defined supercells is absent on 26 April, I'd believe it. In the Knupp et al. (2013) reference, they talk about tornadoes associated with a quasi-linear convective system (QLCS) for the early morning tornadoes in Alabama on 27 April (prior to 12 UTC, so associated with the 26 April convective day) and discrete supercell tornado formation for the afternoon/evening of 27 April. The 26th April storms are much less likely to be supercellular than the 27th April storms. This is reflected in the damage ratings associated with the >10 km path tornadoes for the two days. [EF 26th 27th; EF0 4 3; EF1 20 19; EF2 17 12; EF3 5 15; EF4 0 11; EF5 0 4] The 1240 UTC 27 April tornado in Georgia is associated with the end of 26 April outbreak. There are 4 tornadoes in Alabama (1 EF0, 3 EF1) between 16 and 17 UTC on the 27th that Knupp et al. associate with a small QLCS. The heart of the 27 April outbreak begins after 18 UTC and is unequivocally from supercells, mostly discrete, as opposed to the QLCSs of the 26th (Knupp et al. 2013). The radar loops are available at http://www.srh.noaa.gov/bmx/?n=event_04272011*

[Reply to HB 3]: We accept your comments. We now associate the scatter in **Figure 4b** with tornadoes generated by a number of supercell thunderstorms. We associate the linear texture in **Figure 4a** with a small number of tornadic cells associated with a quasi-linear convective system, and a reference to Knupp *et al.* (2013). We appreciate your clarification and hope that our revisions our consistent with your interpretation.

*[HB 4] p. 5, lines 23-29: The 4 April 2011 event was not supercellular. It was a derecho event with some embedded tornadoes (http://www.spc.noaa.gov/misc/AbtDerechos/casepages/apr042011page.htm). It appears that the technique proposed here with straight lines on the spatial/temporal lag chart is identifying tornado events associated with linear convection, not supercells.*

[Reply to first part of HB 4 (see reply to HB 5 for second part of HB 4)]: We accept that the 4 April 2011 outbreak event was a derecho and have modified our discussion (including a reference) appropriately.

*[HB 5] Figure 5: A single supercell is not an appropriate model for a tornado outbreak such as 27 April 2011 where multiple supercells were producing tornadoes. I've done a simple exercise with 4 supercells on parallel paths. Each produces a tornado once per hour 100 units to the east and 10 units to the north of the previous tornado. The 2nd (3rd) [4th] supercell is 50 (100) [150] units north of the first. When I calculate the lag d and lag t, rather than getting a straight line, I get the plot below. I'm confident that if I made a slightly more sophisticated model with the tornadoes not occurring at the exact same time and not due north of each other, and with not identical forward speeds, the plot would fill in much like Figure 4b. It's probable that, if you created this plot and looked at tornadoes associated with a single supercell separately, you'd see a series of lines. This requires identifying the individual supercell (radar loops would help) or cases such as Palm Sunday 1965, the Superoutbreak in 1974, and May 3 1999 have already had that done. For 27 April, a plot of the tornadoes from Mississippi to northeast Georgia associated with the storm that produced the Tuscaloosa tornado would be illustrative.*

[Figure]

**Fig. 1.** Simple model-4 supercells

[Reply to 2nd part of 4 and 5]: To further address the questions on our modelling, we have carried out a model calculation similar to yours to illustrate conditions for random vs. linear behaviour in the spatial-temporal lag domain. We plan to include this modified discussion (and new figure, see below) in our revised manuscript. We consider a quasi-linear vertical (north-south, y) 'squall' line moving to the east at constant velocity $v = 80$ km hr$^{-1}$, over a 800 km × 800 km region, and a 10 hr period (for tornado touchdowns). Tornadic cells are distributed along the near-linear squall line with an approximate spacing $\Delta y$. Tornadoes are assumed to touch down at equally spaced time intervals (plus some noise $\varepsilon$ that we introduce) $\Delta t + \varepsilon$. The ratio $\Delta y/\Delta t$ defines a characteristic velocity. Our hypothesis is that the non-dimensional velocity ratio $B = (\Delta y)/(v\Delta t)$ defines the behaviour of the system. If $B$ is large ($B > 1$), quasi-linear behaviour is observed in the spatial-temporal lag domain. If $B$ is small ($B < 1$), quasi-random behaviour is observed in the spatial-temporal lag domain. In **Figures I** and **II**, we give two examples. In **Figure Ia** we consider four tornadic cells so that $\Delta y \approx (800$ km$)/4 = 200$ km and ten tornado touch downs from each cell so that $\Delta t \approx 1$ hr. The non-dimensional parameter $B = (\Delta y)/(v\Delta t) = (200$ km$)/[(80$ km hr$^{-1})(1$ hr$)] = 2.5$. In **Figure Ib** we consider ten tornadic cells so that $\Delta y \approx (800$ km$)/10 = 80$ km and four tornado touch downs from each cell so that $\Delta t \approx 2.5$ hr. The non-dimensional parameter $B = (80$ km$)/[(80$ km hr$^{-1})(2.5$ hr$)] = 0.4$.

[Figure]

**Figure I**. Two scenarios for 40 tornadoes in an 800 km x 800 km region over a time period of 10 hr. (a) Four parallel supercells moving at about 80 km hr$^{-1}$ with ten tornadoes each. (b) Ten parallel supercells moving at about 80 km hr$^{-1}$ with four tornadoes each.

[Figure]

**Figure II**. Spatial-temporal lag diagrams for the two scenarios given in **Figure I**.

In **Figures Ia** and **IIa** we give the tornado touchdown locations and times, and in **Figures Ib** and **IIb** we give the spatial-temporal lag results. For Scenario 1, In **Figure IIa**, with *B* = 2.5, we obtain quasi-linear behaviour in the spatial-temporal lag domain, which we consider here further. Refer to the four tornadic lines from top to bottom in **Figure Ia** as A (ten tornadoes along *y* = 725 km), B (*y* = 475 km) C (*y* = 300 km), D (*y* = 90 km). In **Figure IIa**, the spatial-temporal lag domain, the linear correlation passing through the origin results from lags within each of the four tornadic lines A, B, C and D. The next higher set of correlations in the spatial-temporal lag domain in **Figure IIa** (starting at about *d* = 200 km and $\tau$ = 0 hr), is a set of three lines adjacent to each other, and are the result of correlations between tornadic lines A and B, B and C, and C and D. Similarly, the two sets of adjacent lines in **Figure IIa** (starting at about *d* = 400 km and $\tau$ = 0 hr) are the results of correlations between tornadic lines A and C, and B and D. And, finally, the single line in **Figure IIa** (starting at about *d* = 600 km and $\tau$ = 0 hr) is the result of correlation between the tornadic lines A and D in **Figure Ia**. The model we consider is idealized, but we believe it illustrates conditions favourable for linear features (i.e., 26 April 2011) vs. more random features (i.e., 27 April 2011).

*[HB additional comment in separate file] As an example from real data, I went back and got the starting locations for the >10 km tornadoes on 3 May 1999 in the NWSFO Norman forecast area (http://www.srh.noaa.gov/oun/?n=events-19990503-stormdata). I created the plot as in the paper. Combinations associated with a single supercell are shown in red and the combinations of tornadoes from different supercells are shown in black. The individual supercell tornadoes come very close to being on straight lines, although the difference in storm motions leads to slightly different lines between the supercells. The between supercell combinations are scattered around.*

[Figure]

[Reply to HB additional comment]: We thank Harold Brooks for calling our attention to the 3 May 1999 tornado outbreak, and also thank him for additional communications regarding this event, by e-mail correspondence. We have, as did Harold Brooks, applied our technique to this outbreak and the results are given in **Figure III**, which we will introduce in our revised manuscript, along with corresponding text. In **Figure IIIa**, the locations are given as well as the published association with supercell storms. We have calculated (**Figure IIIb**) the spatial temporal lags for (i) all tornadoes associated with storms in that region during the 3 May 1999 (18 tornadoes, Storms A, B, D, E, G, H), (ii) just those tornadoes associated with Storm B (4 tornadoes), and (iii) just those tornadoes associated with Storm D (5 tornadoes). For all storms, the spatial-temporal lags show considerable lag, but in the cases for the individual storms, linear correlations are obtained that pass through the origin, with corresponding velocities 43 (Storm D) and 38 (Storm B) km hr$^{-1}$. We believe the addition of this event, and corresponding discussion, will be a strong addition to our manuscript.

[Figure]

**Figure III**. 3 May 1999 Tornado outbreak. (a) Touchdown locations of 18 severe ($L \geq 10$ km) tornadoes, with storms A, B, D, E, G, H responsible for the tornadoes identified. (b) Spatial-temporal lag correlations between the touchdowns for (i) all 18 tornadoes given in (a), (ii) four tornadoes from Storm B, (iii) five tornadoes from Storm C. Also given are the best-fit linear lines for spatial-temporal lags for both Storms B and D.